# Comparative genomics explains the evolutionary success of reef-forming corals

Debashish Bhattacharya[1,2]*, Shobhit Agrawal[3], Manuel Aranda[3], Sebastian Baumgarten[3], Mahdi Belcaid[4], Jeana L Drake[5], Douglas Erwin[6], Sylvian Foret[7,8], Ruth D Gates[4], David F Gruber[9,10], Bishoy Kamel[11], Michael P Lesser[12], Oren Levy[13], Yi Jin Liew[3], Matthew MacManes[14], Tali Mass[5,15], Monica Medina[11], Shaadi Mehr[9,16], Eli Meyer[17], Dana C Price[18], Hollie M Putnam[4], Huan Qiu[1], Chuya Shinzato[19], Eiichi Shoguchi[19], Alexander J Stokes[20,21], Sylvie Tambutté[22], Dan Tchernov[15], Christian R Voolstra[3], Nicole Wagner[1], Charles W Walker[14], Andreas PM Weber[23], Virginia Weis[17], Ehud Zelzion[1], Didier Zoccola[22], Paul G Falkowski[5,24]*

[1]Department of Ecology, Evolution and Natural Resources, Rutgers University, New Brunswick, United States; [2]Department of Marine and Coastal Sciences, Rutgers University, New Brunswick, United States; [3]Red Sea Research Center, Biological and Environmental Sciences and Engineering Division, King Abdullah University of Science and Technology (KAUST), Thuwal, Saudi Arabia; [4]Hawaii Institute of Marine Biology, Kaneohe, United States; [5]Environmental Biophysics and Molecular Ecology Program, Department of Marine and Coastal Sciences, Rutgers University, New Brunswick, United States; [6]Smithsonian Institution, National Museum of Natural History, Washington, United States; [7]ARC Centre of Excellence for Coral Reef Studies, James Cook University, Townsville, Australia; [8]Research School of Biology, Australian National University, Canberra, Australia; [9]American Museum of Natural History, Sackler Institute for Comparative Genomics, New York, United States; [10]Department of Natural Sciences, City University of New York, Baruch College and The Graduate Center, New York, United States; [11]Department of Biology, Mueller Lab, Penn State University, University Park, United States; [12]School of Marine Science and Ocean Engineering, University of New Hampshire, Durham, United States; [13]The Mina and Everard Goodman Faculty of Life Sciences, Bar-Ilan University, Ramat Gam, Israel; [14]Department of Molecular, Cellular and Biomedical Sciences, University of New Hampshire, Durham, United States; [15]Marine Biology Department, The Leon H. Charney School of Marine Sciences, University of Haifa, Mt. Carmel, Israel; [16]Biological Science Department, State University of New York, College at Old Westbury, New York, United States; [17]Department of Integrative Biology, Oregon State University, Corvallis, United States; [18]Department of Plant Biology and Pathology, Rutgers University, New Brunswick, United States; [19]Marine Genomics Unit, Okinawa Institute of Science and Technology Graduate University, Okinawa, Japan; [20]Laboratory of Experimental Medicine and Department of Cell and Molecular Biology, John A. Burns School of Medicine, Honolulu, United States; [21]Chaminade University, Honolulu, United States; [22]Centre Scientifique de Monaco, Quai Antoine Ier, Monaco; [23]Institute of Plant Biochemistry, Heinrich-Heine-Universität, Düsseldorf, Germany; [24]Department of Earth and Planetary Sciences, Rutgers University, New Jersey, United States

*For correspondence:
d.bhattacharya@rutgers.edu (DB);
falko@marine.rutgers.edu (PGF)

**Abstract** Transcriptome and genome data from twenty stony coral species and a selection of reference bilaterians were studied to elucidate coral evolutionary history. We identified genes that encode the proteins responsible for the precipitation and aggregation of the aragonite skeleton on which the organisms live, and revealed a network of environmental sensors that coordinate responses of the host animals to temperature, light, and pH. Furthermore, we describe a variety of stress-related pathways, including apoptotic pathways that allow the host animals to detoxify reactive oxygen and nitrogen species that are generated by their intracellular photosynthetic symbionts, and determine the fate of corals under environmental stress. Some of these genes arose through horizontal gene transfer and comprise at least 0.2% of the animal gene inventory. Our analysis elucidates the evolutionary strategies that have allowed symbiotic corals to adapt and thrive for hundreds of millions of years.

## Introduction

Reef-building stony corals (Scleractinia) and their cnidarian ancestors have created many thousands of square kilometers of biomineralized marine habitat in shallow tropical seas since their extensive radiation in the Middle Triassic period ~240 million years ago (Ma) (*Veron, 1995*). Coral reefs provide a significant source of ecosystem-based services (*Moberg and Folke, 1999*) that stabilize coastlines and provide habitat for an astounding variety of flora and fauna (*Connell, 1978*). To better understand the evolutionary strategies underpinning the evolutionary success of reef-building corals, we analyzed genomic and transcriptomic information from twenty stony corals that contain intracellular photosynthetic dinoflagellate symbionts of the genus *Symbiodinium* (https://comparative.reefgenomics.org/) (*Figure 1*, and *Figure 1—source data 1*). In addition, bilaterian reference gene sets and genomes from other cnidarians, ctenophores, sponges, a choanozoan, and a placozoan were integrated into our analysis. The comprehensive reference database used for our study included 501,991 translated protein sequences from 20 coral species, 98,458 proteins from five other cnidarians such as sea anemone and sea fan, and 91,744 proteins from seven basal marine metazoan lineages such as sponges and ctenophores. These publicly available genomic and transcriptomic data, which showed large disparities in terms of numbers of predicted protein sequences per species were 'cleaned' of contaminants and poor quality data with the use of stringent filters and selection criteria (see Materials and methods). This procedure resulted in a reasonably comprehensive coverage of corals (i.e., 20 species in total, 11 robust clade species including 2 genomes, 9 complex clade species including 1 genome) with and average of 21,657 protein sequences per species. Given the challenges associated with inferring conclusions based on the absence of genes (in particular when analyzing transcriptomic data), our approach focused on identifying ortholog groups present in different taxonomic categories to reach conclusions about genes associated with coral specific traits. This analysis yielded a set of 2485 'root' orthologs, 613 'Non-Cnidaria' orthologs, 462 'Cnidaria' orthologs, 1436 'Anthozoa' orthologs, 1810 'Hexacorallia' orthologs, 172 'A' orthologs, 4751 'Scleractinia' orthologs, 1588 'complex coral' orthologs, and 6,970 'robust coral' orthologs (available at http://comparative.reefgenomics.org/). These orthologs were analyzed to address four major issues in coral evolution: 1) the basis of aragonite exoskeletal accretion that results in reef formation; 2) environmental sensing mechanisms of the cnidarian host; 3) evolution of the machinery necessary to accommodate the physiological risks as well as the benefits associated with the photosynthetic algal symbionts that create a hyperoxic environment when exposed to light; and 4) given the rich microbial flora associated with the coral holobiont (*Fernando et al., 2015*), the contribution of horizontal gene transfer (HGT) to coral evolution. Here we examine novel insights gained in each of these key areas.

## Results

Relying on conserved proteins as queries in BLAST searches against our genomic database, we identified major components of the coral biomineralization toolkit and reconstructed their evolutionary origins using standard phylogenetic methods (see Material and methods). These results are presented in the Discussion section below and summarized in *Figures 2A* and *3*. We also identified

**eLife digest** For millions of years, reef-building stony corals have created extensive habitats for numerous marine plants and animals in shallow tropical seas. Stony corals consist of many small, tentacled animals called polyps. These polyps secrete a mineral called aragonite to create the reef – an external 'skeleton' that supports and protects the corals.

Photosynthesizing algae live inside the cells of stony corals, and each species depends on the other to survive. The algae produce the coral's main source of food, although they also produce some waste products that can harm the coral if they build up inside cells. If the oceans become warmer and more acidic, the coral are more likely to become stressed and expel the algae from their cells in a process known as coral bleaching. This makes the coral more likely to die or become diseased. Corals have survived previous periods of ocean warming, although it is not known how they evolved to do so.

The evolutionary history of an organism can be traced by studying its genome – its complete set of DNA – and the RNA molecules encoded by these genes. Bhattacharya et al. performed this analysis for twenty stony coral species, and compared the resulting genome and RNA sequences with the genomes of other related marine organisms, such as sea anemones and sponges. In particular, Bhattacharya et al. examined "ortholog" groups of genes, which are present in different species and evolved from a common ancestral gene. This analysis identified the genes in the corals that encode the proteins responsible for constructing the aragonite skeleton. The coral genome also encodes a network of environmental sensors that coordinate how the polyps respond to temperature, light and acidity.

Bhattacharya et al. also uncovered a variety of stress-related pathways, including those that detoxify the polyps of the damaging molecules generated by algae, and the pathways that enable the polyps to adapt to environmental stress. Many of these genes were recruited from other species in a process known as horizontal gene transfer.

The oceans are expected to become warmer and more acidic in the coming centuries. Provided that humans do not physically destroy the corals' habitats, the evidence found by Bhattacharya et al. suggests that the genome of the corals contains the diversity that will allow them to adapt to these new conditions.

major components of the ion trafficking systems in human genomes, and searched for their orthologs in corals (*Figure 2B* and *Figure 2—source data 1*). Finally, using the approach described above, we identified stress response genes in corals and other cnidarians (listed in *Supplementary file 1*).

To elucidate the impact of foreign gene acquisition in coral evolution, we estimated the extent of HGT in the genomic data using a conservative phylogenomic approach (see Materials and methods). This procedure was followed by localization of key HGT candidates to genomic contigs to validate their provenance (*Figure 4*). Using the *A. digitifera* and *Seriatopora* sp. proteomes independently as queries resulted in 13,256 and 19,700 alignments of which 21 and 41, respectively (i.e., in *A. digitifera, Seriatopora* sp.), supported HGT (62/32,956 trees = 0.2%). After accounting for gene duplicates and redundancy between the trees, we discovered 41 unique instances of foreign gene acquisition from bacteria and algae (*Table 1*). Of these candidates, 28 genes were present in the anthozoan common ancestor (i.e., were shared with anemone and/or sea fan) and 13 were specific to corals. In all cases, the HGT-derived genes were shared by multiple anthozoan species and the phylogenies of these genes were largely consistent with the reference tree shown in *Figure 1*.

## Discussion

### Coral biomineralization

The most obvious feature of corals over geological time is their fossilized calcium carbonate skeletons, of which the original mineral component is aragonite. It has been hypothesized for many years that the precipitation of aragonite is catalyzed by and organized on an extracellular organic matrix

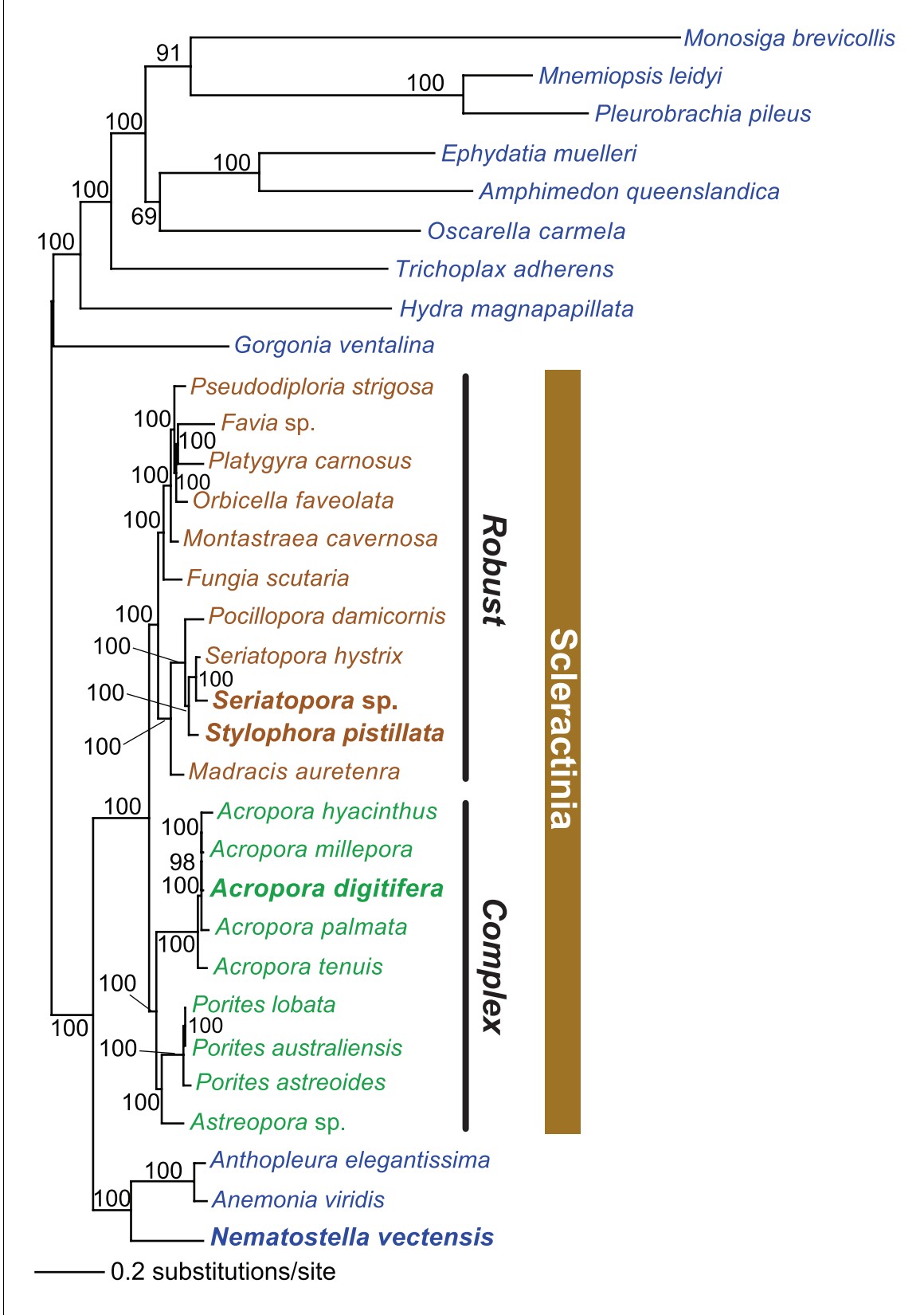

**Figure 1.** Multigene maximum likelihood (RAxML) tree inferred from an alignment of 391 orthologs (63,901 aligned amino acid positions) distributed among complete genome (boldface taxon names) and genomic data from 20 coral species and 12 outgroups. The PROTGAMMALGF evolutionary

*Figure 1 continued*

model was used to infer the tree with branch support estimated with 100 bootstrap replicates. Robust and complex corals are shown in brown and green text, respectively, and non-coral metazoan species are shown in blue text.

The following source data is available for figure 1:

**Source data 1.** Coral genomic data compiled in this study and their attributes.

containing a suite of proteins, lipids, and polysaccharides (*Mann, 2001*; *Watanabe et al., 2003*). This process is precisely controlled and occurs in the calcifying fluid lined by the ectodermal calicoblastic cells that initiate and control the precipitation reaction. Four major components are involved in the process and will be described below: a source of inorganic carbon, a source of calcium ions, proteins that catalyze the nucleation reaction, and proteins and other organic molecules that organize the crystals to form macroscopic structures (*Figure 2A*). In this figure, only the transcellular pathway at the level of the calicoblastic cells is shown. Calcium presumably enters the cells *via* a calcium channel (*Zoccola et al., 1999*) and exits through a calcium ATPase which is proposed to remove protons from the site of calcification (*Zoccola et al., 2004*). Whereas part of the dissolved inorganic carbon (DIC) can enter the cells *via* a bicarbonate transporter (*Furla et al., 2000*), the major source of DIC comes from metabolic $CO_2$, which either diffuses out of the cells through the membranes or is intracellularly converted into $HCO_3^-$ due to a favorable pH (*Venn et al., 2009*), a reaction which is accelerated by carbonic anhydrases (*Bertucci et al., 2013*). This bicarbonate can then exit the cells *via* a bicarbonate transporter (*Zoccola et al., 2015*). At the site of calcification carbonic anhydrases can also play a role in the kinetics of the interconversion between carbon dioxide and bicarbonate (*Bertucci et al., 2013*) according to the extracellular pH (*Venn et al., 2011*). The organic matrix which plays different roles in the biological precipitation of carbonates, comprises a set of proteins including CARPs (*Mass et al., 2013*; ), collagens (*Drake et al., 2013*), galaxins (*Fukuda et al., 2003*), and carbonic anhydrase related proteins (*Drake et al., 2013*).

More broadly, inorganic carbon in seawater in the upper ocean is approximately 2 mM with 95% in the form of bicarbonate ions and is delivered to the site of calcification from an internal pool within the host animal (*Erez, 1978*; *Furla et al., 2000*). This happens either by diffusion of $CO_2$ or by active transport of $HCO_3^-$ following $CO_2$ hydration (*Tambutté et al., 1996*). The hydration reaction is catalyzed by an intracellular carbonic anhydrase (CA) (*Bertucci et al., 2013*). To help facilitate calcification, calicoblastic cells concentrate dissolved inorganic carbon (DIC) in the calcifying fluid (*Allison et al., 2014*). Analysis of our genome data shows two distinct families of bicarbonate anion transporters (BATs) in the coral *Stylophora pistillata* (*Zoccola et al., 2015*). Three isoforms belong to the SLC26 family (*Figure 2—figure supplement 1*) and 5 isoforms belong to the SLC4 family (*Figure 2—figure supplement 2*). One isoform, SLC4γ, is restricted to scleractinians and is only expressed in the calicoblastic cells (*Zoccola et al., 2015*), strongly suggesting that this protein plays a key role in calcification. This bicarbonate transporter could either supply DIC at the site of calcification, or aid in pH regulation in addition to a calcium ATPase (see below). Furthermore, the two BAT gene families are split along phylogenetic lines between the robust and complex clades of scleractinians.

The concentration of calcium ions in seawater is 10 mM, with these ions being actively transported by the calicoblastic cells to the calcifying fluid (*Tambutté et al., 1996*). Radiocalcium ($^{45}Ca$) and inhibitor studies demonstrate that calcium entry in calicoblastic cells by facilitated diffusion is dependent on voltage-gated calcium channels (*Tambutté et al., 1996*). Based on their alpha 1 subunit ($Ca_v\alpha1$) these channels can be phylogenetically divided into three groups. Specific inhibitors (dihydropyridines) strongly suggest that these channels belong to the voltage-dependent L-type family $Ca_v$ 1 and have been characterized at the molecular level and localized by immunohistochemistry in the calicoblastic cells (*Zoccola et al., 1999*). We constructed a phylogeny of the alpha 1 subunit of all types of $Ca_v$ (*Figure 2—figure supplement 3*) and found orthologs in most of the datasets used here, as previously shown for the actinarian *Nematostella vectensis* and the scleractinian *Acropora millepora* (*Moran and Zakon, 2014*). Calcium efflux from the calicoblastic cells to calcifying fluid likely occurs through a plasma membrane calcium ATPase (Ca-ATPase) (*Zoccola et al., 2004*). This enzyme is also responsible for removing protons and increasing pH in the calcifying fluid

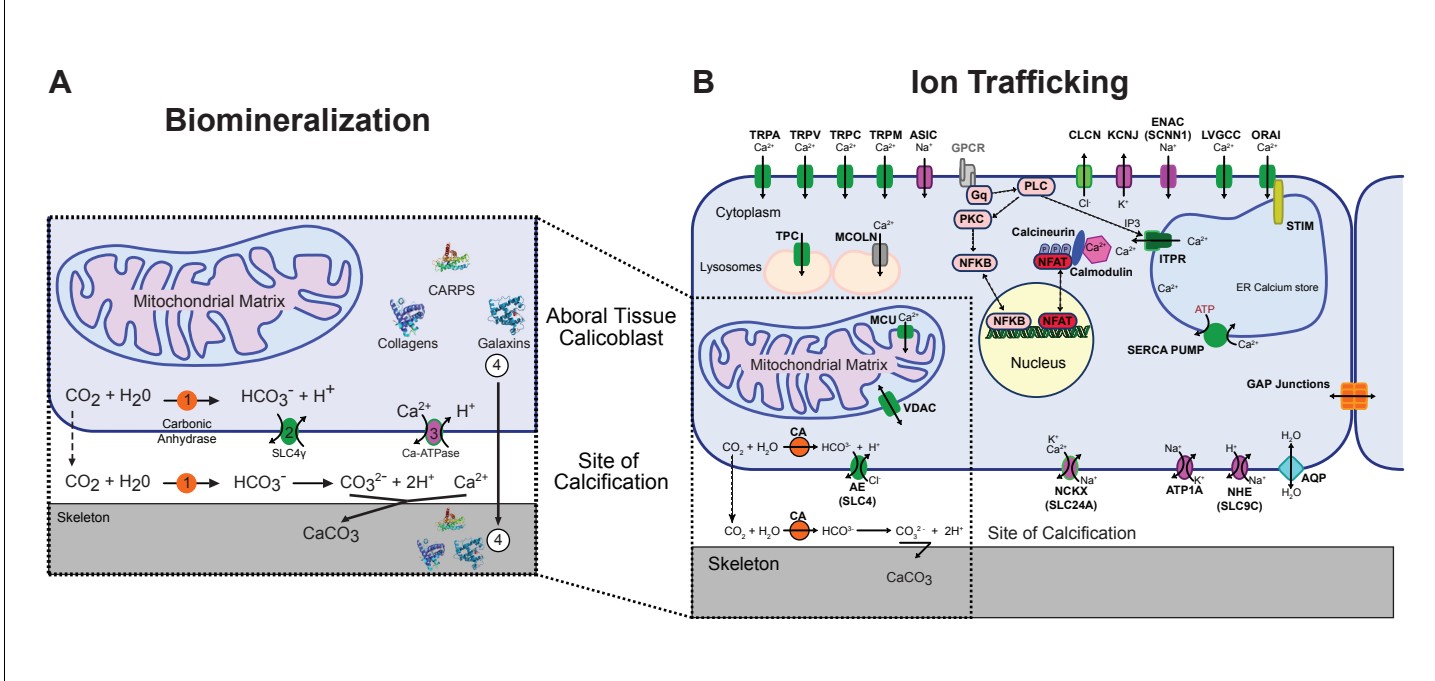

**Figure 2.** The mechanism of (A) coral biomineralization based on data from physiological and molecular approaches and (B) the major components of the human ion trafficking system that were identified in the coral genomic data (*Figure 2—source data 1* for details). Here, in (A) Biomineralization, 1 = carbonic anhydrases (orange); 2 = bicarbonate transporter (green); 3 = calcium-ATPase (purple); 4 = organic matrix proteins (shown as protein structures).

The following source data and figure supplements are available for figure 2:

**Source data 1.** Major components of the human ion trafficking system identified in the coral genomic data.
**Figure supplement 1.** Bayesian consensus trees of SLC26.
**Figure supplement 2.** Bayesian consensus trees of SLC4.
**Figure supplement 3.** Bayesian consensus trees of $Ca_v$.
**Figure supplement 4.** Bayesian consensus trees of coral and outgroup Ca-ATPase proteins.
**Figure supplement 5.** Evolution of CARPs and other coral acid-rich proteins.
**Figure supplement 6.** Scatter plot of isoelectric points of collagens from *Seriatopora*, *Stylophora*, *Nematostella*, and *Crassostrea gigas*.
**Figure supplement 7.** Maximum likelihood (ML) trees of galaxin and amgalaxin.

in order to increase the aragonite saturation state to promote calcification (*Zoccola et al., 2004*; *Venn et al., 2011*; *Davy et al., 2012*). For this enzyme (*Figure 2—figure supplement 4*) as well as for calcium channels and bicarbonate transporters, there is a division between the robust and complex clades of scleractinians.

As described in the two previous paragraphs, for the analysis of the source of inorganic carbon and calcium transport for biomineralization, we focused on the molecules which were previously characterized both by pharmacological and physiological studies in order to link molecular characterization to function. Our data clearly show that transporters such as calcium channels and calcium ATPases and some bicarbonate transporter isoforms are ubiquitously present in the calcifying and non-calcifying cnidarians (scleractinian corals and sea anemones). Based on the genomic analysis of

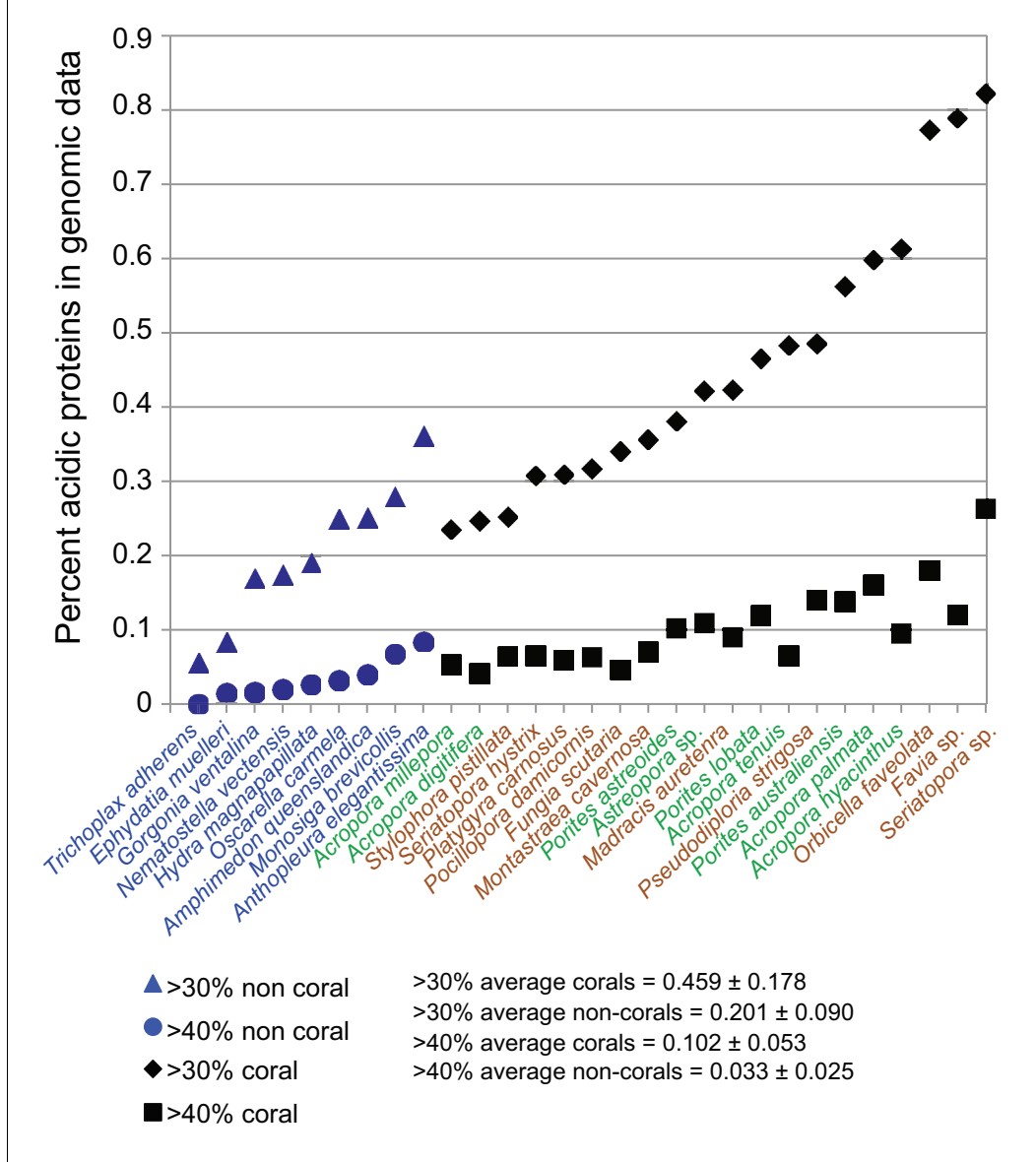

**Figure 3.** Comparison of robust coral (brown text) and complex coral (green text) and non-coral (blue text) genomes with respect to percent of encoded proteins that contain either >30% or >40% negatively charged amino acid residues (i.e., aspartic acid [D] and glutamic acid [E]). The average composition and standard deviation of D + E is shown for the two cut-offs of these estimates. On average, corals contain >2-fold more acidic residues than non-corals. This acidification of the coral proteome is postulated to result from the origin of biomineralization in this lineage.

bicarbonate transporters families in two scleractinian corals and one sea anemone, *Zoccola et al. (2015)* observed that one isoform of the bicarbonate transporter family SLC4γ, was restricted to scleractinians. The current transcriptomic analysis of calcifying and non-calcifying cnidarian species confirms this result, which underlines the role of this transporter in biomineralization. Additional studies are however needed to localize this transporter in different coral species and to determine whether, as for *S. pistillata*, it is also specifically expressed in the calicoblastic cells. Another important piece of information is that for all the different enzymes and transporters studied, there is generally a division in the phylogenetic tree between the robust and the complex clades of scleractinian corals. This suggests that the different calcification traits observed for the two clades (for example, complex corals have less heavily calcified skeletons than robust corals), are due to differences in the biochemical characteristics of these enzymes and transporters.

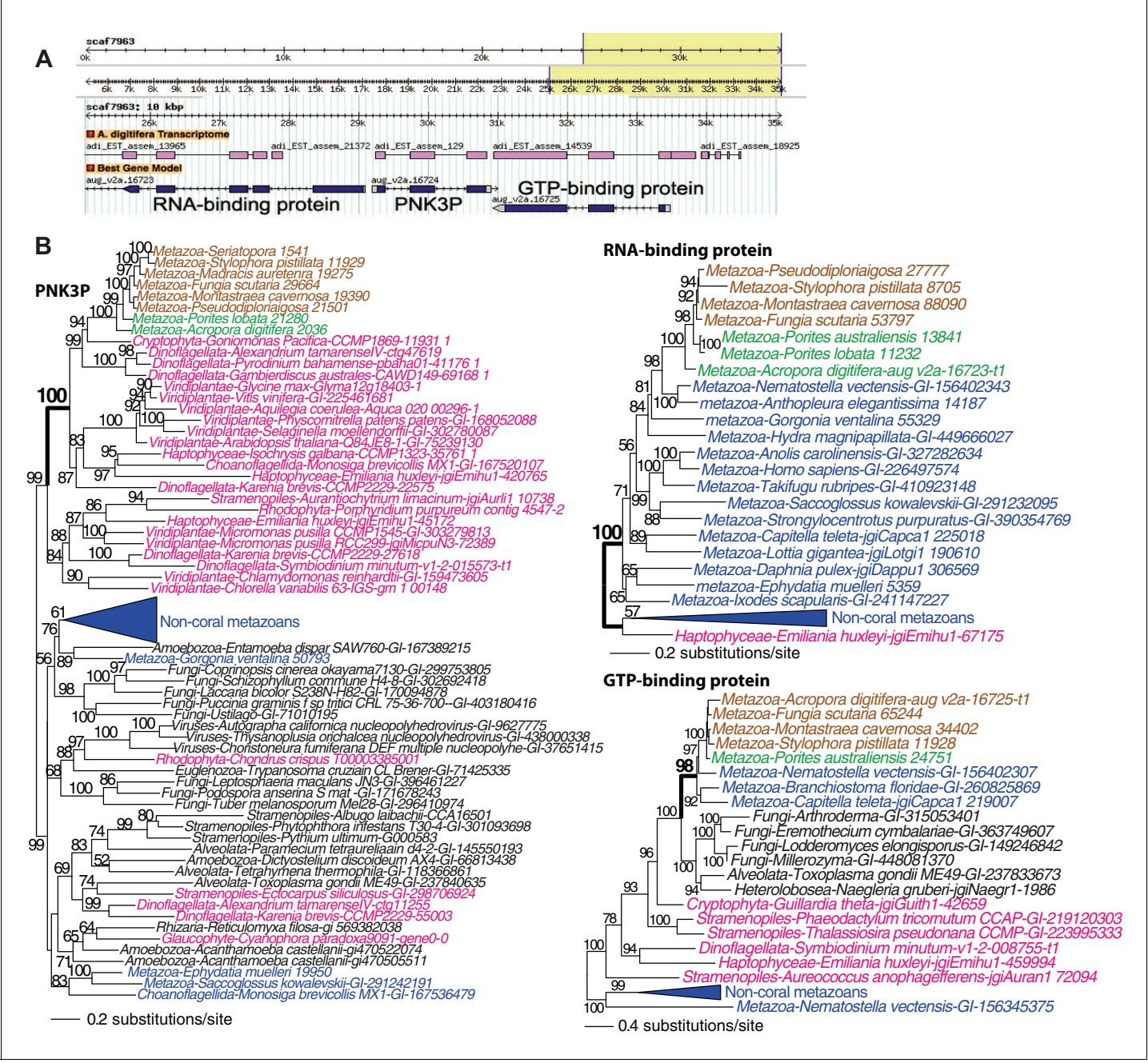

**Figure 4.** Analysis of a genomic region in *Acropora digitifera* that encodes a putative HGT candidate. (**A**) The genome region showing the position of the HGT candidate (PNK3P) and its flanking genes. (**B**) Maximum likelihood trees of PNK3P (polynucleotide kinase 3 phosphatase, pfam08645) domain-containing protein and the proteins (RNA-binding and GTP-binding proteins) encoded by the flanking genes. Robust and complex corals are shown in brown and green text, respectively, and non-coral metazoan and choanoflagellate species are shown in blue text. Photosynthetic lineages, regardless of phylogenetic origin, are shown in magenta text and all other taxa are in black text. GenBank accession (GI) or other identifying numbers are shown for each sequence. The PNK3P domain plays a role in the repair of DNA single-strand breaks by removing single-strand 3'-end-blocking phosphates (*Petrucco et al., 2002*).

The following figure supplements are available for figure 4:

**Figure supplement 1.** Maximum likelihood trees of a DEAD-like helicase and the protein encoded by the flanking gene.

**Figure supplement 2.** Maximum likelihood tree of an exonuclease-endonuclease-phosphatase (EEP) domain-containing protein (**A**), an ATP-dependent endonuclease (**B**), a tyrosyl-DNA phosphodiesterase 2-like protein (**C**), and DNA mismatch repair (MutS-like) protein (**D**).

*Figure 4 continued on next page*

*Figure 4 continued*

**Figure supplement 3.** Maximum likelihood trees of glyoxalase I (or lactoylglutathione lyase) and the proteins encoded by the flanking genes (top image) in *Acropora digitifera*.

**Figure supplement 4.** Maximum likelihood tree of a second glyoxalase I (or lactoylglutathione lyase) and the proteins encoded by the flanking genes (top image) in *Acropora digitifera*.

**Figure supplement 5.** Maximum likelihood tree of an algal-derived short-chain dehydrogenase/reductase (**A**), and a dinoflagellate-derived phosphonoacetaldehyde hydrolase (**B**).

The skeletal structure of corals contains an embedded organic matrix with a set of proteins that have a high proportion of aspartic and glutamic acids (*Mitterer, 1978*; *Weiner, 1979*; *Mann, 2001*; *Weiner and Dove, 2003*; *Gotliv et al., 2005*). These coral acid-rich proteins (CARPs) (Mass et al., 2013) show sequence similarity across Scleractinia (*Drake et al., 2014*) and have functional analogs across the biomineralizing tree of life (*Gorski, 1992*; *Sarashina and Endo, 2001*; *Kawasaki et al., 2009*). CARPs contain >28% aspartic and glutamic acids and have isoelectric points less than pH 5 (*Table 1* in *Mass et al., 2013*). Each of these proteins can individually catalyze the precipitation of calcium carbonate in vitro in natural seawater (*Mass et al., 2013*), hence, they appear to be responsible for initiating biomineralization. Our results show that the average composition of aspartic and glutamic acids in scleractinian corals is >2-fold higher than in 12 non-calcifying invertebrates, with no obvious difference between the robust and complex clades of scleractinians (*Figure 3*). Moreover, phylogenetic analysis reveals that four CARP genes (CARPs 2–5) are widely distributed among scleractinians, suggesting they are derived from homologs present in non-calcifying anthozoans. Extensive duplication of genes encoding CARPs predated the split of robust and complex corals can be seen for CARPs 3–5 (*Figure 2—figure supplement 5*), whereas CARP 2 appears to be unique to robust corals. A previous hypothesis that CARP 1 resulted from a gene (domain) fusion (*Mass et al., 2013*) is supported by these extensive genome data. CARP 1 is derived from a reticulocalbin-like gene present in all metazoans that underwent the fusion of an acidic N-terminal domain, resulting in a modular gene that is found only in corals (*Figure 2—figure supplement 5*). Our data suggest that the enrichment of highly negatively charged proteins is a major distinguishing feature of stony corals.

At the nanoscale, the biological precipitation of aragonite crystals is insufficient to form the highly organized, stable macrostructures that characterize corals. The crystals are organized by a series of proteins that act as 'glues'. One of these protein families, found in the skeletons of corals is collagen (*Jackson et al., 2010*; *Drake et al., 2013*). In basal invertebrates, there are three families of collagen (fibrillar, multiplexins, and type IV) that are also present in vertebrates. Other than their structural function, collagens play an important role in the regulation of cell-cell adhesion, differentiation, and wound healing (*Heino et al., 2009*). Collagens in the alpha IV subfamily have been identified in the organic matrix of coral skeletons (*Ramos-Silva et al., 2013*; *Drake et al., 2013*). Alpha IV collagens form networks of fibers that are an important component of the extracellular matrix. Using the complete genome data from *S. pistillata* and *Seriatopora* sp., we identified 230 and 208 predicted open reading frames (ORFs), respectively, that contained a collagen Pfam domain. Of these, 52 *S. pistillata* proteins contain an extracellular secretion signal, in comparison to 17 from *Seriatopora* sp. By plotting the isoelectric point (IP) of the secreted collagens from both corals we identified four acid-rich collagens in *Seriatopora* sp. and five in *S. pistillata* that have an IP < 7 (*Figure 2—figure supplement 6*). This analysis strongly suggests that these collagens play a critical role in tethering aragonite crystals in coral skeletons similar to their role in bone formation (*Nudelman et al., 2010*).

In addition to collagens, stony corals secrete a variety of other adhesion proteins into the calcifying milieu (*Ramos-Silva et al., 2013*; *Drake et al., 2013*). These include cadherins, which facilitate cell-cell or cell-substrate adhesion, vitellogenin, and zonadhesin proteins. As part of the biomineralization toolkit, these proteins bind the calicoblastic cells to the newly formed skeleton and may assist in the binding of CARPs to other functional proteins. Interestingly, the first protein sequenced from coral skeleton, galaxin, is neither acidic nor calcium binding, and its function remains unknown

**Table 1.** The list of non-redundant anthozoan genes derived *via* HGT.

| No. | Ancestor | Genes | Protein products | Support | Source(s) |
|---|---|---|---|---|---|
| 1 | Coral | *A. digitifera*_2036 | PNK3P | 100 | CA |
| 2 | Coral | *A. digitifera*_8849 | SDR | 100 | CA |
| 3 | Coral | *Seriatopora*_31861 | DEAD-like helicase | 100 | Bact |
| 4 | Coral | *Seriatopora*_16594 | glyoxalase | 100 | CA |
| 5 | Coral | *Seriatopora*_17147 | acyl- dehydrogenase | 100 | Bact |
| 6 | Coral | *Seriatopora*_17703 | carbonic anhydrase | 96 | Dino |
| 7 | Coral | *Seriatopora*_19477 | fatty acid or sphingolipid desaturase | 100 | CA |
| 8 | Coral | *Seriatopora*_3957 | atpase domain-containing protein | 100 | Bact |
| 9 | Coral | *Seriatopora*_7060 | sam domain-containing protein | 100 | Bact |
| 10 | Coral | *Seriatopora*_7928 | atp phosphoribosyltransferase | 100 | CA/Fungi |
| 11 | Coral | *Seriatopora*_8296 | glyoxalase | 98 | Bact |
| 12 | Coral | *Seriatopora*_22596 | 2-alkenal reductase | 92 | Bact |
| 13 | Coral | *Seriatopora*_28321 | histidinol-phosphate aminotransferase | 96 | Unclear |
| 14 | Anthozoa | *A. digitifera*_418 | duf718 domain protein | 100 | CA |
| 15 | Anthozoa | *A. digitifera*_15871 | peptidase s49 | 96 | Algae/Bact |
| 16 | Anthozoa | *A. digitifera*_14520 | predicted protein | 100 | CA/Bact |
| 17 | Anthozoa | *A. digitifera*_7178 | rok family protein/fructokinase | 93 | Red algae |
| 18 | Anthozoa | *A. digitifera*_10592 | Phospholipid methyltransferase | 100 | CA/Viri |
| 19 | Anthozoa | *A. digitifera*_13390 | predicted protein | 100 | Bact |
| 20 | Anthozoa | *A. digitifera*_313 | malate synthase | 98 | CA/Bact |
| 21 | Anthozoa | *A. digitifera*_1537 | hypothetical protein | 100 | Bact |
| 22 | Anthozoa | *A. digitifera*_13577 | gamma-glutamyltranspeptidase 1-like | 100 | Unclear |
| 23 | Anthozoa | *A. digitifera*_5099 | Isocitrate lyase (ICL) | 100 | Bact |
| 24 | Anthozoa | *A. digitifera*_13467 | uncharacterized iron-regulated protein | 100 | CA |
| 25 | Anthozoa | *A. digitifera*_6866 | 3-dehydroquinate synthase | 98 | CA |
| 26 | Anthozoa | *A. digitifera*_11675 | intein c-terminal splicing region protein | 100 | Bact |
| 27 | Anthozoa | *Seriatopora*_10994 | penicillin amidase | 100 | Bact |
| 28 | Anthozoa | *Seriatopora*_14009 | nucleoside phosphorylase-like protein | 100 | Bact |
| 29 | Anthozoa | *Seriatopora*_14494 | phosphonoacetaldehyde hydrolase | 100 | Dino |
| 30 | Anthozoa | *Seriatopora*_15303 | exonuclease-endonuclease-phosphatase | 99 | CA/Viri |
| 31 | Anthozoa | *Seriatopora*_15772 | fmn-dependent nadh-azoreductase | 99 | Dino |
| 32 | Anthozoa | *Seriatopora*_19888 | had family hydrolase | 97 | Algae/Bact |
| 33 | Anthozoa | *Seriatopora*_20039 | chitodextrinase domain protein | 92 | Dino |
| 34 | Anthozoa | *Seriatopora*_20146 | glutamate dehydrogenase | 100 | CA/Bact |
| 35 | Anthozoa | *Seriatopora*_20479 | thif family protein | 100 | Bact |
| 36 | Anthozoa | *Seriatopora*_21195 | ATP-dependent endonuclease | 100 | Dino |
| 37 | Anthozoa | *Seriatopora*_8585 | chitodextrinase domain protein | 92 | Bact |
| 38 | Anthozoa | *Seriatopora*_24047 | aminotransferase | 100 | Bact |
| 39 | Anthozoa | *Seriatopora*_25961 | d-alanine ligase | 99 | Bact |
| 40 | Anthozoa | *Seriatopora*_26478 | quercetin 3-o-methyltransferase | 100 | Viri |
| 41 | Anthozoa | *Seriatopora*_29443 | diaminopimelate decarboxylase | 100 | CA |

Bact: Bacteria; CA: chlorophyll *c*-containing algae; Dino: dinoflagellates; Viri: Viridiplantae.

(*Fukuda et al., 2003*). Originally sequenced from *Galaxea fascicularis*, but more recently identified in the *A. millepora* skeleton, galaxin is a 30–40 kDa glycosylated protein with a signal peptide, suggesting it is secreted (*Fukuda et al., 2003*; *Ramos-Silva et al., 2013*). The primary sequence contains ~20 paired cysteine (CC) residues. Usherin, found in vertebrates has a similar high number of paired cysteine motifs (*Baux et al., 2007*) and binds type IV collagens (*Bhattacharya et al., 2004*), suggesting a potential role for this galaxin. Galaxin was originally suggested to be coral-specific (*Fukuda et al., 2003*), however, galaxin-like proteins are found in non-calcifying taxa outside Cnidaria (e.g., *Sanchez et al., 2007*; *Heath-Heckman et al., 2014*). Therefore, it has been proposed that the precursor to modern coral galaxin homologs was recruited as a biomineralization protein when Scleractinia diverged from non-biomineralizing taxa during the Triassic (*Foret et al., 2010*). Our sequence analysis supports this hypothesis, suggesting that not only is coral galaxin derived from a common ancestor with non-calcifying metazoans, but that it is polyphyletic within corals (*Figure 2—figure supplement 7*), and independently recruited for a role in biomineralization multiple times in coral evolution. The first evidence for stony corals occurs in the Triassic and fossil evidence shows a rapid proliferation of taxa (reviewed by *Stanley, 2003*); this was also a time of 'aragonite seas' when geochemical conditions were favorable to the formation and evolution of aragonitic coral skeletons (*Stanley and Hardie, 1998*).

A second type of galaxin, amgalaxin, has an N-terminal acidic domain that precedes the galaxin domain (*Reyes-Bermudez et al., 2009*). However, unlike galaxin, amgalaxin appears to function only in the early larval stages of biomineralization and has not been observed in the coral skeleton (*Reyes-Bermudez et al., 2009*; *Ramos-Silva et al., 2013*). This pattern is similar to the mollusk and coral proteins nacrein and CARP1 (see above), in which an acidic domain is fused to an existing gene (*Miyamoto et al., 1996*; *Mass et al., 2013*). Unlike galaxin, the acidic portion of amgalaxin appears to be limited to corals (*Figure 2—figure supplement 7*). This result suggests that the attachment of an acidic region to galaxin is unique to stony corals and that amgalaxin, like CARP1, emerged from a gene fusion event.

## Environmental and stress response systems

Corals typically produce planktonic or 'crawl-away larvae' that calcify when they settle on an appropriate benthic substrate, and have thereby effectively determined their future physical environment for the life of the organism. Hence, habitat selection is one of the most critical elements in the survival and success of individual corals. To help accommodate variations in habitat on time scales varying from hours to years, corals have evolved a suite of environmental sensing and response systems. One of the most critical environmental cues for coral success is light (*Dubinsky and Falkowski, 2011*). Stony corals use diel periodicity and light sensing capabilities as cues for spawning, feeding, and orienting the polyps. Perhaps not surprisingly, the host animal has genes encoding a circadian clock. However, the light sensing signal cascades in zooxanthellate corals are particularly complex because of their symbiotic relationship with dinoflagellates, which also have a circadian clock. Coral environmental response genes are coupled to the dinoflagellate circadian clock, anticipating changes in the intracellular milieu such as the coral tissue becoming hyperoxic due to zooxanthellate photosynthesis and near-hypoxic at night due to host and symbiont respiration. Numerous chaperones such as heat shock protein (hsp) 40, hsp70, hsp90, grp94, hsp90b1, calreticulin, and protein disulfide isomerase are 'hard-wired' to this photosynthesis/respiration clock and the high level of synchrony of circadian transcription of chaperones and antioxidant genes reflects the diurnal preparedness of the coral to the consequences of oxidative protein damage imposed by photosynthesis of the algal symbionts (*Levy et al., 2011*). Symbiosis also indirectly imposes diurnal gene expression fluctuations, most likely *via* the hypoxia inducible factor (HIF) system. In a wide array of animals, glycolytic enzymes are regulated by HIF1-alpha transcription factor, a clear ortholog of which is present in the 20 coral genomic datasets. The HIF system is unique to animals, and HIF itself is a target of calpain-mediated degradation in vertebrates. Calpains are $Ca^{2+}$-dependent regulatory proteases and corals linkage of calpain expression to the HIF system potentially enables them to utilize cellular calcium levels to modulate expression of other HIF targets when hypoxia dominates (*Levy et al., 2011*).

The casein kinase I (CK1) family consists of serine/threonine protein kinases that are key regulators of circadian timing in bilaterian animals, fungi, and green algae (*van Ooijen et al., 2013*). CK1-like encoding genes are found in most corals and were suggested to be components of the coral

circadian gene network along with CLOCK, GSK₃B/Sgg, and CSNK1D (*Vize, 2009*). The proteins ADCI, GNAQ, GNAS, GNB1, CREB1, and NOS1 are related to G-protein coupled receptor signaling and can act on neuropeptide/GPCR-coupled signaling mechanisms. This is consistent with neurohormones playing a role in synchronized spawning events in tropical abalone (*York et al., 2012*) and in coral larvae settlement (*Grasso et al., 2011*). Other proteins such as PPEF1 and GRIN1 respond to light stimulus, whereas MTNR1A and MTNR1B are melatonin receptors, whereas PRKAA2 is a protein kinase that responds to peptide hormone stimulus and is responsive to circadian rhythms. The circadian processes are impacted by catabolic process; i.e., *S. pistillata* glycolysis is controlled by ARNT and HIF1-alpha that provide feedback that affects the circadian loop. Surprisingly, BLASTp analysis of the 20 coral genomic datasets did not turn up the Period gene as reported in other cnidarians. Therefore, the core circadian clock architecture of the negative feedback loop in basal metazoans such as corals may differ significantly from animal lineages that diverge after corals.

Although fluxes of calcium and bicarbonate ions into the calicoblastic space are part of the biomineralization system, these and other ion pumps also generate electrochemical gradients that allow stony corals to sense the environment and initiate complex and specific signaling cascades (*Hille, 1986*). This ion trafficking landscape and downstream signaling components are comprised of channels, transporters, exchangers, pumps, second messenger generators, and transcriptional response elements. Many of these ion transporters act as direct physicochemical sensors providing intra-cellular and intra-organismal regulation and the critical linkage between external environmental changes and cytoplasmic and organellar events, cascades and transcriptional regulation. We identified major components of the ion trafficking systems in human genomes, and searched for their orthologs in corals (*Figure 2B*). Ion channel sensors such as the transient receptor potential (TRP) channels (TRPA, TRPV, TRPM, TRPC) (*Ramsey et al., 2006*; *Nilius and Szallasi, 2014*) and acid sensing channels (ASICs) are present in corals (*Krishtal, 2015*). Most of these are either direct, or indirect, physicochemical sensors of environmental parameters such as temperature, pH and oxygen tension. Organelle ion regulators such as two-pore channels (TPCN) (*Wang et al., 2012*; *Horton et al., 2015*), mucolipin (MCOLN) are also present and are thought to maintain intraorganellar pH and ion environments. In summary, most, if not all of these components sense environmental changes and implement signaling cascades that lead to the activation of specific transcriptional programs that allow the organism to physiologically respond to environmental signals.

## Impacts of the environment on the symbiotic life history of corals

Symbiotic corals thrive in oligotrophic tropical and subtropical seas in large part because their intracellular, symbiotic dinoflagellates provide a significant portion of their photosynthesis-derived fixed carbon to the host animal. However, this benefit comes with significant costs. The ecological stability of the symbiotic association is dependent on it being stable in the face of environmental extremes. This symbiosis has been widely described as living close to the upper extremes of thermal tolerance that, when exceeded, leads to a cascade of cellular events resulting in 'coral bleaching', whereby corals lose their symbiotic algae and consequently one of their main sources of carbon (*Lesser, 2006*; *2011*). Other environmental extremes can lead to coral bleaching including exposure to ultraviolet radiation (UVR) and ocean acidification (*Lesser, 2004*; *Hoegh-Guldberg et al., 2007*). Proximately, in this cascade of events, many physiological studies on bleaching in corals and other symbiotic cnidarians have shown that photosynthetically produced hyperoxic conditions act synergistically with solar radiation, especially UVR, and thermal stress to produce reactive oxygen species (ROS) and reactive nitrogen species (RNS) in both host tissues and *Symbiodinium* sp. beyond their capacity to quench these toxic products (*Lesser, 2006*; *2011*). Ultimately a series of fairly well described stress response events involving cell cycle arrest and apoptosis, in both the algal symbionts and host, appear to be responsible for the massive expulsion of dinoflagellates from the host, and ultimately, host mortality if the environmental insult is severe enough or of prolonged duration (*Lesser, 2006*; *2011*)

Therefore, the ecological stability of the symbiotic association in zooxanthellate corals requires increased stability in the face of environmental extremes. Previous coral genomic studies have identified genes involved in the stress response of cnidarians (*Shinzato et al., 2011*), but here we show that corals contain highly conserved genes involved in oxidative stress, DNA repair, the cell cycle and apoptosis (*Supplementary file 1*). For instance we identify both the extrinsic and intrinsic apoptotic pathways, characteristic of many vertebrates including humans. These genes are not derived

by HGT in the Cnidaria, because of their presence in poriferans and other sister taxa (see HGT discussion below). Corals exposed to oxidative stress, or UVR, accumulate DNA damage, whereby cell cycle arrest occurs and cell repair is initiated (*Lesser and Farrell, 2004*). If DNA damage is too severe, then a cellular cascade leading to genetically programmed cell death by apoptosis occurs via an intrinsic, or mitochondrial, pathway. Whereas the intrinsic pathway is considered a response to stress (e.g., thermal stress), the extrinsic, or death-receptor pathway is a cellular process by which cell to cell communication activates apoptosis via ligand binding to cell surface receptors, as in the well described immunological response to cancer cells or pathogens. Genes present in cnidarians and active in the vertebrate intrinsic DNA damage induced apoptotic pathway include: ATM, p53 (and many of its important regulator proteins and transcriptional products), Hausp, Bax, Bcl-2, AIF, cytochrome C, APAF1, procaspase 9, procaspase 3, ICAD and CAD (*Supplementary file 1*). The activity of these genes in cnidarians comprises the cellular machinery necessary to accomplish the following: mitochondrial catastrophe, apoptosome formation, breakdown of the nuclear pores, intra-nuclear DNA disassembly and flipping of phosphotidylserine from the inner to the outer leaflet of the plasma membrane that in humans permits macrophage recognition of apoptotic cells. In addition, we identified a complete nitric oxide synthase (NOS; EC 1.14.13.39) in corals. This gene is derived from a metazoan ancestor and is thought to play a key role in the stress response that leads to breakdown of the symbiosis and coral bleaching (*Trapido-Rosenthal et al., 2005*; *Hawkins et al., 2013*). Another significant finding of our analysis of multiple taxa is that Bid (BH3; Bcl-2 domain of homology 3), the only protein that allows the extrinsic and intrinsic pathways in vertebrates to directly communicate with each other, is not present in the coral data. Previous research on apoptosis in invertebrates, particularly on the intrinsic pathway, demonstrated the conserved nature of the molecular machinery in ancestral metazoans (*Bender et al., 2012*). Cnidarians encode all the genes for both pathways known to be expressed and active in vertebrates, but appear to lack the ability to communicate between them. This function is mediated by p53, the gatekeeper for cell growth and division, through Bid in vertebrates (*Sax et al., 2002*) that is present in 20 of 25 cnidarian datasets examined here. The antiquity of the intrinsic pathway is striking and along with the recent demonstration of a functional extrinsic pathway in cnidarians (*Quistad et al., 2014*) reveals the importance of these apoptotic pathways in metazoan evolution. Interestingly, tumor necrosis factor (TNF), an essential mediator of the extrinsic death-receptor pathway, was present in only 7 of the 32 datasets examined in this study (*Supplementary file 1*). Lastly, the presence of the major genes in the human extrinsic and intrinsic pathways suggests that cnidarians may be a potential model system for studying transcriptionally induced apoptosis, when compared to *Caenorhabditis elegans* and *Drosophila melanogaster.* In these latter animal models, the available functional data indicate that genes in the cellular senescence, DNA editing, and repair pathways that are governed by the transcriptional activation domain (TAD) of p53 are only 2% (*D. melanogaster*) and 33% (*C. elegans*) conserved when compared to human p53 (*Walker et al., 2011*). This result suggests limited control of somatic cell apoptosis in these organisms perhaps because their adult somatic cells do not divide by mitosis.

## Contribution of horizontal gene transfer to coral evolution

The primary function of the HGT candidates we identified in stony corals is to extend the existing stress related pathways in these animals. These foreign genes encode proteins that provide protection from UVR and stress from reactive species (*Banaszak and Lesser, 2009*; *Nesa et al., 2012*). It has already been reported that corals and sea anemones acquired a pathway that produces photoprotective mycosporine amino acids that absorbs UVR (*Shinzato et al., 2011*). Our results show additions to the DNA repair pathway, including a polynucleotide kinase 3-phosphatase (PNK3P) of algal origin (*Figure 4*) and a DEAD-like helicase of bacterial origin (*Figure 4—figure supplement 1*). These two genes are flanked by eukaryotic or coral-specific genes in their respective contigs in the draft genome of *A. digitifera* (*Figure 4* and *Figure 4—figure supplement 1*). Two DNA repair genes that were transferred from algal sources were found in the anthozoan ancestor. These encode an exonuclease-endonuclease-phosphatase (EEP) domain-containing protein and an ATP-dependent endonuclease (*Figure 4—figure supplement 2*). Furthermore, two DNA repair genes are shared between Anthozoa and sponges or choanoflagellates, but are missing from a large diversity of Bilateria; these encode a tyrosyl-DNA phosphodiesterase 2-like protein and a DNA mismatch repair (MutS-like) protein (*Figure 4—figure supplement 2*). Our results fit in well with the so-called Public

Goods Hypothesis that posits important genetic resources, such as mechanisms of DNA repair, are distributed widely among taxa via both vertical and horizontal evolution (*McInerney et al., 2011*).

Protection against reactive species in corals, in addition to the multiple homologs we found with antioxidant functions such as superoxide dismutase (*Supplementary file 1*), is provided by two genes derived via HGT that encode glyoxalase I. One of these has an algal (*Figure 4—figure supplement 3*) and the other a bacterial provenance (*Figure 4—figure supplement 4*). Interestingly, the latter gene is physically located between a DNA repair gene (encoding RAD51) and a tRNA modification gene on scaffold 2777 in the *A. digitifera* draft assembly (*Figure 4—figure supplement 4*). Glyoxalase I belongs to a system that carries out the detoxification of reactive carbonyls (RC), such as highly cytotoxic methylglyoxal, produced by sugar metabolism and the Calvin cycle (*Shimakawa et al., 2014*). Methylglyoxal production in plastids increases with light intensity (*Takagi et al., 2014*). Another gene encoding a putative RC scavenger (*Shimakawa et al., 2014*) is short-chain dehydrogenase/reductase (SDR) that was derived in corals from an algal source (*Figure 4—figure supplement 5*). Other alga-derived HGTs were from species containing plastids of red algal secondary endosymbiotic origin (i.e., chlorophyll *c*-containing lineages such as stramenopiles) (*Table 1*). Given the coral-*Symbiodinium* symbiosis, it is also notable that several of the HGT candidates appear to be derived from dinoflagellates (e.g., *Figure 4—figure supplement 5*). The gene contribution from chlorophyll *c*-containing lineages suggests a long history of interaction between these algae and the anthozoan lineage.

## Conclusions

Cnidarians enter the fossil record about 545 Ma in the latest Ediacaran Period and have been an important component of marine ecosystems throughout the Phanaerozoic, surviving five major mass extinctions and many smaller biotic crises. Although reefs have often disappeared during each of these events, various coral clades have persisted. Our analysis of a subset of coralliform cnidarians, the symbiotic Scleractinia, reveals how their genomic information has provided the basis for adapting to changes in ocean temperature and pH, while maintaining the ability to calcify. This is significant because scleractinians survived throughout the Cenozoic despite atmospheric $CO_2$ levels reaching 800 ppm 50–34 Ma, and tropical sea temperatures of 30°–34°C from 45 to 55 Ma (*Norris et al., 2013*). This interval coincides with a reef gap, but reefs were quickly re-established thereafter. The resilience of corals in the face of extraordinary changes in ocean conditions clearly bespeaks a gene inventory that is highly adaptive as exemplified by the diversification of CARPs and genes recruited through HGT. Human activity has the potential to further reduce the abundance of these organisms in coming decades; indeed, there is compelling evidence of human destruction of corals worldwide. However, the diverse genetic repertoire of these organisms will potentially allow them to survive the expected changes in thermal structure and pH in the coming centuries (*Stolarski et al., 2011*), assuming that their populations and habitats are not physically destroyed by humans.

## Materials and methods

### Analysis of genome data and construction of coral tree of life

Coral genomic and transcriptome data compiled in this study are summarized in *Figure 1—source data 1*. All data were filtered to remove assembled contigs <300 bp. ORFs were predicted with TransDecoder (*Haas et al., 2013*) yielding amino acid sequences. Protein duplicates were subsequently removed with CD-HIT (*Fu et al., 2012*). With regard to coral sequence datasets, potential contaminant sequences from the algal symbiont, *Symbiodinium* were removed with script psytrans. py (https://github.com/sylvainforet/psytrans) using training sets retrieved from *Symbiodinium microadriaticum* (*Baumgarten et al., 2013*) and *Acropora digitifera* (*Shinzato et al., 2011*). Successful separation of coral and algal sequences was validated by GC-content plots that showed a clear bimodal data distribution (results not shown). Filtered sequence data were searched against SwissProt (*Boutet et al., 2007*), TrEMBL (*Bairoch and Apweiler, 2000*), NCBI nr databases using BLASTp (Basic Local Alignment Search Tool, *e*-value cut-off = 1e-03) (*Altschul et al., 1990*) and retaining annotations from databases in this order. BLAST2GO (*Conesa et al., 2005*) was queried to provide GO annotations, and KEGG (*Kanehisa and Goto, 2000*), Pfam (*Bateman et al., 2002*),

InterProScan (*Zdobnov and Apweiler, 2001*) were searched to further annotated gene sets. Filtered and annotated genomic and transcriptomic data are available at comparative.reefgenomics.org.

Orthologs were identified using InParanoid (*Ostlund et al., 2010*) on pairwise BLASTp (*e*-value cutoff = 1e-05) yielding a list of pairwise orthologs that was subsequently queried with QuickPara-noid (http://pl.postech.ac.kr/QuickParanoid/) for automatic ortholog clustering among multiple spe-cies. QuickParanoid input files were filtered according to the following rules: A) Only orthologs sets were retained with a confidence score of 1, and B) Pairwise comparisons were retained if only one sequence is present in each of the two involved species. To make more robust inferences based on transcriptomic data, we filtered our ortholog dataset such that any ortholog from a given phyloge-netic grouping (i.e., robust corals, complex corals, Scleractinia, Actiniaria, Hexacorallia, Anthozoa, Cnidaria, non-cnidarian, root) was considered to be an ortholog in this group if it was present in this group and absent in all other groups. The QuickParanoid-derived ortholog clusters were sorted into the following categories based on the constituent taxa and known species tree (*Figure 1*): 1.) 2,485 'root' orthologs, 2.) 613 'Non-Cnidaria' orthologs, 3.) 462 'Cnidaria' orthologs, 4.) 1436 'Anthozoa' orthologs, 5.) 1,810 'Hexacorallia' orthologs, 6.) 172 'Actiniaria' orthologs, 7.) 4,751 'Scleractinia' orthologs, 8.) 1,588 'complex coral' orthologs, and 9.) 6,970 'robust coral' orthologs (available at http://comparative.reefgenomics.org). For phylogenetic tree building, we selected 'root' orthologs that were present in at least 50% of the species of any lineage (i.e. Root, Non-Cnidarian, Cnidarian, Anthozoa, Hexacorallia, Actiniaria, Scleractinia, Complex corals, Robust corals) yielding 391 distinct orthologs over 7970 sequences. Orthologs were aligned individually on the protein level via MAFFT (*Katoh and Standley, 2013*) in 'LINSI' mode. The resulting alignments were concatenated and then trimmed with TrimAl in the automated mode (-*automated*) (*Capella-Gutierrez et al., 2009*). The resulting alignment (63,901 amino acids) was used for phylogenetic tree building with RAxML (*Sta-matakis, 2014*) under PROTGAMMALGF model with 100 bootstrap replicates for the estimation of branch supports (-T 32 -f a -x 1234 -p 1234 -N 100 -m PROTGAMMALGF).

## Analysis of ion transport

Human ionome protein reference sequences were identified and downloaded from Genbank at NCBI. Using BLASTStation-Local64 (v1.4, TM software, Inc, Arcadia, CA 91007, USA), a coral protein database was generated. This contained all protein sequences available from the reefgenomics web-site (http://comparative.reefgenomics.org/). The human ionome protein sequences were then used as queries to search (Basic Local Alignment Search Tool, BLAST) against this local database using BLASTp (no filter, Expect: 10; Word Size 3; Matrix: BLOSUM63; Gap Costs: Existence 11 extension 1) using BLASTStation-Local64. The resulting matching coral proteins were saved in multi-FASTA for-mat files, and then re-BLASTed against the NCBI Refseq protein database (*Pruitt et al., 2012*) lim-ited to human-only proteins (taxid:9606) on the NCBI BLAST webportal (algorithm BLASTp, default parameters; Expect: 10; Word Size 3; Matrix: BLOSUM62; Gap Costs: Existence 11 extension 1) (*Camacho et al., 2009*). The results were viewed for each coral protein from the input file, and a summary was generated, indicating which human protein was identified as a top hit, and in which coral species it was found. The coral multi-FASTA file was copied and annotated manually with the gene symbols of the human protein identified. If a protein coral sequence was not identified as the original human protein sequence, it was deleted, if other gene family members were identified this information was also annotated, and entered into the summary table. These multi-FASTA files were then stored for future analysis (e.g., generating phylogenetic trees). The results from the coral to human BLASTp alignments were also stored.

## Analysis of horizontal gene transfer

Protein sequences in RefSeq (version 58) were downloaded from NCBI FTP site (ftp://ftp.ncbi.nlm. nih.gov/refseq/). When sequences were available from more than one (sub) species in a genus (e.g., *Arabidopsis thaliana* and *A. lyrata* in the genus *Arabidopsis*), the species (e.g., *A. thaliana*) with larg-est number of sequence were retained, whereas others (e.g., *A. lyrata*) were all removed. This data-set was combined with algal sequences collected from Cryptophyta [*Guillardia theta* (*Curtis et al., 2012*)], Haptophyceae [*Emiliania huxleyi* (*Read et al., 2013*)], Rhizaria [*Bigelowiella natans* (*Curtis et al., 2012*) and *Reticulomyxa filose* (*Glockner et al., 2014*)], Stramenopiles [*Nannochlorop-sis gaditana* (*Radakovits et al., 2012*) and *Aureococcus anophagefferens* (*Gobler et al., 2011*)] and

dinoflagellates [*Alexandrium tamarense* (*Keeling et al., 2014*), *Karenia brevis* (*Keeling et al., 2014*), *Karlodinium micrum* (*Keeling et al., 2014*), *Symbiodinium minutum* (*Shoguchi et al., 2013*)], Glauco-phyte [*Cyanophora paradoxa* (*Price et al., 2012*)], Viridiplantae [*Bathycoccus prasinos* (*Moreau et al., 2012*), *Chlorella variabilis* (*Blanc et al., 2010*), *Coccomyxa subellipsoidea* (*Blanc et al., 2012*), *Micromonas pusilla* (*Worden et al., 2009*), *Glycine max* (*Schmutz et al., 2010*)] and all red algal sequences collected in the previous study (*Qiu et al., 2015*). We further clustered similar sequences (sequence identity ≥85%) among taxa from each order (e.g., Brassicales or Primates), retained the longest sequence and removed all other related sequences in the same cluster using CD-HIT version 4.5.4 (*Li and Godzik, 2006*). This non-redundant database, combined with protein sequences derived from three coral genomes (*Acropora digitifera* and *Seriatopora* sp. and *Stylophora pistillata*) was designated as 'Ref58+Coral' database.

The protein sequences from *A. digitifera* and *Seriatopora* sp. genomes were used as query to search against the 'Ref58+Coral' database using BLASTp (*e*-value cut-off = 1e-05). Up to 1000 top hits (query-hit identity ≥27.5%) were recorded. These hits were sorted according to query-hit identity in a descending order among those with query-hit alignment length (≥120 amino acids). Hit sequences were then retrieved from the queried database with no more than three sequences for each order and no more than 12 sequences for each phylum. The resulting sequences were aligned using MUSCLE version 3.8.31 (*Edgar, 2004*) under default settings and trimmed using TrimAl version 1.2 (*Capella-Gutierrez et al., 2009*) in an automated mode (-*automated1*). Alignment positions with ≥50% gap were discarded. We removed sequence alignments with <80 amino acid sites and those with <10 sequences. The remaining alignments were used for phylogenetic tree building using FastTree version 2.1.7 (*Price et al., 2010*) under the defaulting settings (except that WAG model was used instead of JTT model). The resulting trees were parsed to search for coral sequences that were nested within metazoan sequences with ≥0.9 local support values estimated using the Shimo-daira-Hasegawa test (*Shimodaira and Hasegawa, 1999*) using in-house tools. All such coral sequences were considered to represent metazoan host genes and were discarded from downstream analyses.

We conducted a second run of phylogenomic analysis using an expanded database comprising 'Ref58+Coral' database and all metazoan sequences collected in this study (http://comparative.reef-genomics.org/datasets.html). The analyses were performed following the aforementioned procedure except that phylogenetic trees were constructed using RAxML (*Stamatakis, 2014*) under PROTGA-MMALGF model with branch supports estimated using 100 bootstrap replicates. With these RAxML trees, we searched for coral sequences that were nested within non-metazoan sequences (with ≥60% bootstrap support). The resulting phylogenetic trees were manual inspected to identify HGT candidates. HGT cases that were unique to the query species (not shared with any other coral taxa) were discarded. The tree topologies for the resulting candidates were confirmed by re-building the trees using IQtree version 0.96 (*Nguyen et al., 2015*) under the best amino acid substitution model selected by the build-in model-selection function. Branch supports were estimated using ultrafast bootstrap (UFboot) approximation approach (*Minh et al., 2013*) using 1500 bootstrap replicates (-bb 1500). Coral sequences were considered to have a HGT origin if they were nested within non-metazoan sequences with ≥90% UFboot support. When phylogenetic trees derived from the *A. digitifera* data and those derived from *Seriatopora* sp. showed the same HGT event (i.e., an ancient transfer that occurred before the split of these two species), they were manually grouped into a shared non-redundant group. The same was the cases for phylogenetic trees that resulted from recent gene duplications. This process gave rise to 21 *A. digitifera* sequences and 41 *Seriatopora* sp. sequences that represent 41 independent HGTs from non-metazoan sources (*Table 1*).

The key HGT genes involved in stress response were mapped to *A. digitifera* genome browser using the BLAST function therein (http://marinegenomics.oist.jp/acropora_digitifera). The corresponding phylogenetic trees were rebuilt with inclusion of representative sequences (if available) from more algal taxa (*Pyrodinium bahamense* pbaha01, *Gambierdiscus australes* CAWD149, *Goniomonas Pacifica* CCMP1869, *Togula jolla* CCCM725, *Pleurochrysis carterae* CCMP645, *Ceratium fusus* PA161109) that were generated from the Marine Microbial Eukaryote Transcriptome Sequencing Project (*Keeling et al., 2014*). The alignments were carried out using MUSCLE version 3.8.31 (*Edgar, 2004*) followed by manual trimming and curation (e.g., with the removal of highly divergent sequences and redundant sequences from highly sampled groups). The corresponding ML trees

were built using IQtree (*Nguyen et al., 2015*) as aforementioned. The phylogenetic trees for the flanking genes (if any) were generated likewise.

## Acknowledgements

This work was made possible by grants from the National Science Foundation, especially EF-1408097, to PGF, DB, RDG, HMP and TM, which sponsored the workshop. Additional funding was provided by the National Science Foundation through grants EF-1041143/RU 432635 and EF-1416785 awarded to PGF, DB, and TM, respectively. RDG, HMP, and AJS were supported by grants from the National Institutes of Health, NIMHD P20MD006084, the Hawaii Community Foundation, Leahi Fund 13ADVC-60228 and NSF OCE PRF 1323822 and National Science Foundation Experimental Program to Stimulate Competitive Research Hawaii: EPS-0903833. CRV and MA acknowledge funding by the King Abdullah University of Science and Technology (KAUST).

## Additional information

### Competing interests

PGF: Reviewing editor, *eLife.* The other authors declare that no competing interests exist.

### Funding

| Funder | Grant reference number | Author |
| --- | --- | --- |
| National Science Foundation | EF-1416785 | Paul G Falkowski |

This work was made possible by grants from the National Science Foundation, EF-1041143/RU 432635 and EF-1416785 awarded to PGF, DB, and TM, respectively. RDG, HMP, and AJS were supported by grants from the National Institutes of Health, NIMHD P20MD006084, the Hawaii Community Foundation, Leahi Fund 13ADVC-60228 and NSF OCE PRF 1323822 and National Science Foundation Experimental Program to Stimulate Competitive Research Hawaii: EPS-0903833. CRV and MA acknowledge funding by the King Abdullah University of Science and Technology (KAUST).

### Author contributions

DB, Conceived and coordinated the project and helped write the paper; SA, MA, SB, MB, SF, YJL, SM, CRV, Overall coral genome database and website, Comparative genomic analyses and the coral tree of life; JLD, BK, TM, DZ, HMP, ST, Biomineralization pathway; DE, MMe, CS, ES, APMW, VW, Design of the study; RDG, AJS, Ion transporters; DFG, MPL, OL, MMacMa, EM, DT, CWW, Environmental stress response systems; DCP, Bioinformatic analyses; HQ, Horizontal gene transfer analysis; NW, Seriatopora sp. coral genome data production; EZ, Negatively charged amino acids and other bioinformatic analyses; PGF, Conceived and coordinated the project and helped write the paper, Analysis and interpretation of data, Drafting or revising the article

### Author ORCIDs

Bishoy Kamel, http://orcid.org/0000-0003-2934-3827
Matthew MacManes, http://orcid.org/0000-0002-2368-6960
Alexander J Stokes, http://orcid.org/0000-0002-3526-4685
Christian R Voolstra, http://orcid.org/0000-0003-4555-3795
Didier Zoccola, http://orcid.org/0000-0002-1524-8098
Paul G Falkowski, http://orcid.org/0000-0002-2353-1969

## Additional files

### Supplementary files

• Supplementary file 1. Taxonomic compilation and presence/absence in each taxon for genes involved in oxidative stress, DNA repair, cell cycle and apoptosis. The values in parentheses show the number of taxa in which the gene sequence was recovered in the genomic database.

## Major datasets

The following previously published datasets were used:

| Author(s) | Year | Dataset title | Dataset URL | Database, license, and accessibility information |
|---|---|---|---|---|
| Shinzato C, Shoguchi E, Kawashima T, Hamada M, Hisata K, Tanaka M, Fujie M, Fujiwara M, Koyanagi R, Ikuta T, Fujiyama A, Miller DJ, Satoh N | 2011 | Data from: Acropora digitifera genome to understand coral responses to environmental change | http://www.ncbi.nlm.nih.gov/bioproject/PRJDA67425 | Publicly available at NCBI Bioproject (accession no. PRJDA67425) |
| Barshis DJ, Ladner JT, Oliver TA, Seneca FO, Traylor-Knowles N, Palumbi SR | 2013 | Data from: Genomic basis for coral resilience to climate change | http://www.ncbi.nlm.nih.gov/bioproject/PRJNA177515 | Publicly available at NCBI GenBank (accession no. PRJNA177515) |
| Moya A, Huisman L, Ball EE, Hayward DC, Grasso LC, Chua CM, Woo HN, Gattuso JP, Foret S, Miller DJ | 2012 | Data from: Whole transcriptome analysis of the coral Acropora millepora reveals complex responses to CO2-driven acidification during the initiation of calcification | http://www.ncbi.nlm.nih.gov/bioproject/PRJNA149513 | Publicly available at NCBI Bioproject (accession no. PRJNA149513) |
| Srivastava M, Simakov O, Chapman J, Fahey B, Gauthier ME, Mitros T, Richards GS, Conaco C, Dacre M, Hellsten U, Larroux C, Putnam NH, Stanke M, Adamska M, Darling A, Degnan SM, Oakley TH, Plachetzki DC, Zhai Y, Adamski M, Calcino A, Cummins SF, Goodstein DM, Harris C, Jackson DJ, Leys SP, Shu S, Woodcroft BJ, Vervoort M, Kosik KS, Manning G, Degnan BM, Rokhsar DS | 2010 | Data from: The Amphimedon queenslandica genome and the evolution of animal complexity | http://www.ncbi.nlm.nih.gov/bioproject/?term=PRJNA66531 | Publicly available at NCBI BioProject (accession no. PRJNA66531) |
| Sabourault C, Ganot P, Deleury E, Allemand D, Furla P | 2009 | Data from: Comprehensive EST analysis of the symbiotic sea anemone, Anemonia viridis | http://www.ncbi.nlm.nih.gov/nucest/FK719875 | Publicly available at NCBI EST (accession no. FK719875-FK759813) |
| Kitchen SA, Crowder CM, Poole AZ, Weis VM, Meyer E | 2015 | Data from: De Novo Assembly and Characterization of Four Anthozoan (Phylum Cnidaria) Transcriptomes | http://www.ncbi.nlm.nih.gov/bioproject/PRJNA295078 | Publicly available at NCBI BioProject (accession no. PRJNA295078) |
| Stefanik DJ, Lubinski TJ, Granger BR, Byrd AL, Reitzel AM, DeFilippo L, Lorenc A, Finnerty JR | 2014 | Data from: Production of a reference transcriptome and transcriptomic database (EdwardsiellaBase) for the lined sea anemone | http://cnidarians.bu.edu/EdwardBase/cgi-bin/stock.cgi | Publicly available at EdwardsiellaBase (Edwardsiella lineata genomics database) |
| Pooyaei Mehr SF, DeSalle R, Kao HT, Narechania A, Han Z, Tchernov D, Pieribone V, Gruber DF | 2013 | Data from: Transcriptome deep-sequencing and clustering of expressed isoforms from Favia corals | http://www.ncbi.nlm.nih.gov/bioproject/PRJNA176860 | Publicly available at NCBI BioProject (accession no. PRJNA176860) |

| | | | | |
|---|---|---|---|---|
| Burge CA, Mouchka ME, Harvell CD, Roberts S | 2013 | Data from: Immune response of the Caribbean sea fan, Gorgonia ventalina, exposed to an Aplanochytrium parasite as revealed by transcriptome sequencing | http://www.ncbi.nlm.nih.gov/bioproject/PRJNA172986 | Publicly available at NCBI Bioproject (accession no. PRJNA172986) |
| Chapman JA, Kirkness EF, Simakov O, Hampson SE, Mitros T, Weinmaier T, Rattei T, Balasubramanian PG, Borman J, Busam D, Disbennett K, Pfannkoch C, Sumin N, Sutton GG, Viswanathan LD, Walenz B, Goodstein DM, Hellsten U, Kawashima T, Putnam NH, Shu S, Blumberg B, Dana CE, Gee L, Kibler DF, Law L, Lindgens D, Martinez DE, Peng J, Wigge PA, Bertulat B, Guder C, Nakamura Y, Ozbek S, Watanabe H, Khalturin K, Hemmrich G, Franke A, Augustin R, Fraune S, Hayakawa E, Hayakawa S, Hirose M, Hwang JS, Ikeo K, Nishimiya-Fujisawa C, Ogura A, Takahashi T, Steinmetz PR, Zhang X, Aufschnaiter R, Eder MK, Gorny AK, Salvenmoser W, Heimberg AM, Wheeler BM, Peterson KJ, Böttger A, Tischler P, Wolf A, Gojobori T, Remington KA, Strausberg RL, Venter JC, Technau U, Hobmayer B, Bosch TC, Holstein TW, Fujisawa T, Bode HR, David CN, Rokhsar DS, Steele RE | 2010 | Data from: The dynamic genome of Hydra | http://www.ncbi.nlm.nih.gov/bioproject/PRJNA12876 | Publicly available at NCBI Bioproject (accession no. PRJNA12876) |
| Ryan JF, Pang K, Schnitzler CE, Nguyen AD, Moreland RT, Simmons DK, Koch BJ, Francis WR, Havlak P, NISC Comparative Sequencing Program, Smith SA, Putnam NH, Haddock SH, Dunn CW, Wolfsberg TG, Mullikin JC, Martindale MQ, Baxevanis AD | 2013 | Data from: The genome of the ctenophore Mnemiopsis leidyi and its implications for cell type evolution | http://www.ncbi.nlm.nih.gov/bioproject/PRJNA64405 | Publicly available at NCBI Bioproject (accession no. PRJNA64405) |

| | | | | |
|---|---|---|---|---|
| King N, Westbrook MJ, Young SL, Kuo A, Abedin M, Chapman J, Fairclough S, Hellsten U, Isogai Y, Letunic I, Marr M, Pincus D, Putnam N, Rokas A, Wright KJ, Zuzow R, Dirks W, Good M, Goodstein D, Lemons D, Li W, Lyons JB, Morris A, Nichols S, Richter DJ, Salamov A, Sequencing JG, Bork P, Lim WA, Manning G, Miller WT, McGinnis W, Shapiro H, Tjian R, Grigoriev IV, Rokhsar D | 2008 | Data from: The genome of the choanoflagellate Monosiga brevicollis and the origin of metazoans | http://www.ncbi.nlm.nih.gov/bioproject/PRJNA19045 | Publicly available at NCBI Bioproject (accession no. PRJNA19045) |
| Putnam NH, Srivastava M, Hellsten U, Dirks B, Chapman J, Salamov A, Terry A, Shapiro H, Lindquist E, Kapitonov VV, Jurka J, Genikhovich G, Grigoriev IV, Lucas SM, Steele RE, Finnerty JR, Technau U, Martindale MQ, Rokhsar DS | 2007 | Data from: Sea anemone genome reveals ancestral eumetazoan gene repertoire and genomic organization | http://www.ncbi.nlm.nih.gov/bioproject/PRJNA19965 | Publicly available at NCBI Bioproject (accession no. PRJNA19965) |
| Nichols SA, Roberts BW, Richter DJ, Fairclough SR, King N | 2012 | Data from: Origin of metazoan cadherin diversity and the antiquity of the classical cadherin/beta-catenin complex | http://www.ncbi.nlm.nih.gov/bioproject/PRJNA230415 | Publicly available at NCBI Bioproject (accession no. PRJNA230415) |
| Sun J, Chen Q, Lun JC, Xu J, Qiu JW | 2013 | Data from: PcarnBase: development of a transcriptomic database for the brain coral Platygyra carnosus | http://www.ncbi.nlm.nih.gov/bioproject/PRJNA81573 | Publicly available at NCBI Bioproject (accession no. PRJNA81573) |
| Moroz LL, Kocot KM, Citarella MR, Dosung S, Norekian TP, Povolotskaya IS, Grigorenko AP, Dailey C, Berezikov E, Buckley KM, Ptitsyn A, Reshetov D, Mukherjee K, Moroz TP, Bobkova Y, Yu F, Kapitonov VV, Jurka J, Bobkov YV, Swore JJ, Girardo DO, Fodor A, Gusev F, Sanford R, Bruders R, Kittler E, Mills CE, Rast JP, Derelle R, Solovyev VV, Kondrashov FA, Swalla BJ, Sweedler JV, Rogaev EI, Halanych KM, Kohn AB | 2014 | Data from: The ctenophore genome and the evolutionary origins of neural systems | http://www.ncbi.nlm.nih.gov/bioproject/PRJNA213480 | Publicly available at NCBI Bioproject (accession no. PRJNA213480) |
| Traylor-Knowles N, Granger BR, Lubinski TJ, Parikh JR, Garamszegi S, Xia Y, Marto JA, Kaufman L, Finnerty JR | 2011 | Data from: Production of a reference transcriptome and transcriptomic database (PocilloporaBase) for the cauliflower coral, Pocillopora damicornis | http://cnidarians.bu.edu/PocilloporaBase/cgi-bin/pdamdata.cgi | Publicly available at PocilloporaBase (Pocillopora Transcriptomics Database) |

| | | | | |
|---|---|---|---|---|
| Shinzato C, Inoue M, Kusakabe M | 2014 | Data from: A snapshot of a coral | http://www.ncbi.nlm.nih.gov/bioproject/PRJDB731 | Publicly available at NCBI Bioproject (accession no. PRJDB731) |
| Srivastava M, Begovic E, Chapman J, Putnam NH, Hellsten U, Kawashima T, Kuo A, Mitros T, Salamov A, Carpenter ML, Signorovitch AY, Moreno MA, Kamm K, Grimwood J, Schmutz J, Shapiro H, Grigoriev IV, Buss LW, Schierwater B, Dellaporta SL, Rokhsar DS | 2008 | Data from: The Trichoplax genome and the nature of placozoans | http://www.ncbi.nlm.nih.gov/bioproject/PRJNA30931 | Publicly available at NCBI Bioproject (accession no. PRJNA30931) |
| Schwarz JA, Brokstein PB, Voolstra C, Terry AY, Manohar CF, Miller DJ, Szmant AM, Coffroth MA, Medina M | 2008 | Data from: Coral life history and symbiosis: functional genomic resources for two reef building Caribbean corals | http://www.ncbi.nlm.nih.gov/nucest/DR982333 | Publicly available at NCBI EST (accession no. DR982333-DR988505) |
| Schwarz JA, Brokstein PB, Voolstra C, Terry AY, Manohar CF, Miller DJ, Szmant AM, Coffroth MA, Medina M | 2008 | Data from: Coral life history and symbiosis: functional genomic resources for two reef building Caribbean corals | http://www.ncbi.nlm.nih.gov/nucest/EY021828 | Publicly available at NCBI EST (accession no. EY021828-EY031784) |

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
