## [Decision Letter]

Thank you for submitting your work entitled "A comparative genomic approach to explaining the ecological success of reef-forming corals" for consideration by *eLife*. Your article has been reviewed by two peer reviewers, and the evaluation has been overseen by a Reviewing Editor and Ian Baldwin as the Senior Editor.

Thank you also for your patience while waiting for our evaluation, which took more time than usual. Many of the potential reviewers were not available to assess this manuscript. Several of them expressed a real or perceived conflict of interest because of recent or ongoing collaboration, or affiliation, with one or more of the authors. This was perhaps a unique problem, given the many experts and institutions directly involved with this manuscript. After substantial efforts by the Reviewing Editor, we were able to appoint two unbiased expert reviewers. Given the complexity of the paper, these reviewers required substantial additional time to assess the manuscript.

Both reviewers liked the topic and scope of the paper and found it to be of sufficient broad interest for the audience of *eLife*. The Reviewing Editor agrees with this positive general assessment. However, both reviewers also expressed a number of concerns, which need to be addressed.

The reviewers had opportunity to discuss the reviews with one another and the Reviewing Editor has drafted this decision to help you prepare a revised submission.

Following below is the combined set of the major comments.

Summary of the work:

The manuscript by Bhattacharya and colleagues presents a meta-analysis of genomic sequence data from 20 reef-forming corals including 4 complete coral genome sequences, 5 related cnidarians and 7 basal metazoans generated over many years across numerous platform technologies. The authors consolidate this published information and harmonize gene models in order to generate orthologous clusters for downstream phylogenetic comparisons and metabolic reconstruction. The combined dataset contains ~700,000 predicted gene models. Four focal issues guiding this effort relate to how corals form their exoskeletons, how coral sensing mechanisms enable response to environmental conditions, algal symbiosis trade-offs and horizontal gene transfer. These issues are framed against a backdrop of reef building coral persistence over geological time and the ecosystem services that they provide. Overall, the meta-analysis provides insight into coral evolution and reveals some interesting observations related to coral genome flexibility and host-symbiont HGT. The analysis and discussion focuses on the ecological success of corals, their historical survival and prospect for future populations.

The manuscript represents a rigorous analysis and draws substantially on known organismal, cellular and molecular properties of coral species. It is well written. While the work could be considered to be descriptive in nature it will be of substantial value, not only to coral biologists, but those interested in evolutionary mechanisms of metazoans more generally as well as those interested in mechanisms underlying symbiotic relationships. The paper is likely to have reach with a wide audience. The chief value of the work is that it provides a valuable resource on which to build a whole range of future studies that will require full functional analysis – expression patterns etc. before the proposed functions of the large number of genes and families presented here will be fully understood. The work can thus be considered to be ground-breaking in providing the essential comparative background.

Essential major revisions:

Despite its epic scale and scope, several issues come up in reading the manuscript that should be considered from research and readership perspectives. A description of the published datasets in the meta-analysis needs to be more clearly outlined in the Introduction and Abstract including the non-reef building coral genomes. Was there a community effort to coordinate sequencing of specific coral groups? How were differences in assembly and QC dealt with between projects and platforms? The authors mention the use of genomic and transcriptomic datasets but it is unclear how or if transcriptomic information was used to either increase the number of orthologous groups or cross-validate predicted gene models. More emphasis should be placed in the Introduction on describing the coral holobiont from an organismal point of view and contextualizing ecosystem services provided over geological time. This would provide readers with greater clarity on why the four focal issues driving meta-analysis were selected. Such "experimental logic" would focus the manuscript around a series of unifying questions rather than selected bullet points.

The structure of the manuscript is also problematic. By merging the Results and Discussion it becomes challenging to discern where original contributions were made compared to previously published work. In general, there seemed to be too much emphasis on reviewing previous observations, with little description of the results and their significance. There are entire paragraphs in the Results/Discussion section that do not contain any results e.g. the first paragraph of the 'environmental and stress response systems' section. Rewriting to focus on the results would significantly improve the communication of new observations originating from the meta-analysis. This would allow for a more informative discussion of the implications of these observations as they relate to the four focal issues defined by the authors. Alternatively, the authors might consider converting the current manuscript into a review article.

On a more general note the authors might want to consider the overall information esthetic of the manuscript and develop more unified and interpretable figures. For example, the tree in Figure 1 could be color coordinated with the individual gene trees that come later on. Why were some of 32 genomes used in comparison omitted from this starting tree? Integrating the tree with some of the information in Table 1.g. amount of genomic information or protein encoding gene for each species using bubbles or bars could help define the combined data set more clearly and address some of the issues raised above. Figure 2 contains summary information for two of the four focal issues. Why not provide a panel that summarizes observations related to all four? Better still, why not start with a schematic of a coral holobiont and work inward and outward from there? Figure 3 combines tabular and graphical information. Can the authors simplify the message here? Figure 4 uses colors that are unspecified from an information encoding perspective.

To what extent can the authors be certain that all symbiont sequences have been removed from the database? Given the likely diversity of symbionts, can the authors be confident that they have been able to subtract the symbiont genome information based on just a couple of known genomes?

The authors should provide some caveats about the function of many of the genes described. For example, the calcification model refers to the calicoblastic cells but there is no data to assign location to any of the genes described. This does not detract from the value of the manuscript but I think that this limitation should be acknowledged.

It is not really surprising that corals possess environmental sensors in their genomes. This would be true of just about all eukaryote cells. To infer that the presence of sensing components (at the end of the subsection “Environmental and stress response systems”) that are pretty ubiquitous indicates a special propensity to respond to varying environmental signals is not really original. More interesting would be the demonstration of further novel environmental signalling components, as indicated by the analysis of circadian genes and the interesting findings with extrinsic and intrinsic apoptotic pathways. In relation to the latter, a better explanation of intrinsic and extrinsic pathways would add clarity to the Discussion.

---

## [Author Response]

Essential major revisions:

Despite its epic scale and scope, several issues come up in reading the manuscript that should be considered from research and readership perspectives. A description of the published datasets in the meta-analysis needs to be more clearly outlined in the Introduction and Abstract including the non-reef building coral genomes. Was there a community effort to coordinate sequencing of specific coral groups? How were differences in assembly and QC dealt with between projects and platforms? The authors mention the use of genomic and transcriptomic datasets but it is unclear how or if transcriptomic information was used to either increase the number of orthologous groups or cross-validate predicted gene models.

We have added to the revised manuscript a thorough description of the sources (see [Supplementary-material SD1-data]), as well the methods used to generate the database. Where available, the genes or preassembled transcripts were used after a rigorous filtering and quality control step. Similarly, rigorous cutoffs were applied to the ortholog detection step.

In more detail, we removed assembled contigs (not sequence reads!) <300bp in size (which translated to ORFs that are at least 100 amino acids in length) in order to increase the robustness of the annotations and of ortholog assignment. Every effort was made to find a reasonable balance between the sensitivity of our analyses and the comprehensiveness of the sequence data set. The average number of protein sequences per species considered in our analyses is 21,657. Although the variance between datasets is admittedly high (from 2,183 to 66,449 protein sequences) this is an unavoidable consequence of incorporating data from multiple sources and the stringent filtering we applied in order to make the data more reliable and consistent across species.

In our analyses, we focused on genes *present* in the species because we are well aware of the challenges associated with drawing conclusions based on gene absence in transcriptomic data. However, to reduce our reliance on the transcriptomic data (for the majority of species), we conditioned our ortholog dataset so that any ortholog from a given phylogenetic grouping (i.e., robust corals, complex corals, Scleractinia, Actineria, Hexacorallia, Anthozoa, cnidarian, non-cnidarian, root) was considered to be a feature in the group, only if it was present in that group and absent in all other phylogenetic groupings. We are confident that our data represent a reasonably comprehensive coverage of coral genes (i.e., 20 species in total, 11 robust clade species including 2 genomes, 9 complex clade species including 1 genome) and, although we cannot exclude the possibility that some coral genes were most certainly missing in our database, this proportion is likely to be small given our broad taxonomic coverage.

*More emphasis should be placed in the Introduction on describing the coral holobiont from an organismal point of view and contextualizing ecosystem services provided over geological time. This would provide readers with greater clarity on why the four focal issues driving meta-analysis were selected. Such "experimental logic" would focus the manuscript around a series of unifying questions rather than selected bullet points.*We expanded the Abstract and Introduction to provide some additional detail about the data generation and its uses. We hope this change accommodates the reviewer’s comments. As it is clear with further reading, the manuscript provides quite a bit of background about each research aim in the newly created Discussion section. This latter text should provide readers with sufficient perspective about why we chose to pursue this study and its major outcomes.

The structure of the manuscript is also problematic. By merging the Results and Discussion it becomes challenging to discern where original contributions were made compared to previously published work. In general, there seemed to be too much emphasis on reviewing previous observations, with little description of the results and their significance. There are entire paragraphs in the Results/Discussion section that do not contain any results e.g. the first paragraph of the 'environmental and stress response systems' section. Rewriting to focus on the results would significantly improve the communication of new observations originating from the meta-analysis. This would allow for a more informative discussion of the implications of these observations as they relate to the four focal issues defined by the authors. Alternatively, the authors might consider converting the current manuscript into a review article.

The manuscript was reorganized as the reviewers suggest and now has separate Results and Discussion sections.

On a more general note the authors might want to consider the overall information esthetic of the manuscript and develop more unified and interpretable figures. For example, the tree in Figure 1 could be color coordinated with the individual gene trees that come later on. Why were some of 32 genomes used in comparison omitted from this starting tree? Integrating the tree with some of the information in Table 1.g. amount of genomic information or protein encoding gene for each species using bubbles or bars could help define the combined data set more clearly and address some of the issues raised above. Figure 2 contains summary information for two of the four focal issues. Why not provide a panel that summarizes observations related to all four? Better still, why not start with a schematic of a coral holobiont and work inward and outward from there? Figure 3 combines tabular and graphical information. Can the authors simplify the message here? Figure 4 uses colors that are unspecified from an information encoding perspective.

Taxon coloring was made consistent throughout the manuscript, where possible. Figure 1, Figure 3, and 4 now use the same coloring scheme. Other changes to the figures such as adding all analyzed taxa to Figure 1 are described below. We did not change Figure 2, which we feel addresses well two of the major aims, and with the new Discussion section there is now a clearer demarcation between the four research aims.

*To what extent can the authors be certain that all symbiont sequences have been removed from the database? Given the likely diversity of symbionts, can the authors be confident that they have been able to subtract the symbiont genome information based on just a couple of known genomes?*We used the PSyTrans package (https://github.com/sylvainforet/psytrans) to separate sequences of the host species from those of its main symbiont(s) based on Support Vector Machine (SVM) classification. Successful separation of coral host and algal symbiont sequences was checked via GC-content plots of host- and symbiont-sorted sequences that displayed a clear bimodal distribution, as expected for a successful separation. The cross-validation accuracy of the SVM was between 96% and 99%. Moreover, given that our analyses are based on orthologous sequences, it is unlikely that contamination from the same source is present over multiple species/data sets. Given this approach, the probability of finding the same *Symbiodinium* ortholog in all the datasets within a phylogenetic grouping (e.g., robust corals) is likely to be very low.

The authors should provide some caveats about the function of many of the genes described. For example, the calcification model refers to the calicoblastic cells but there is no data to assign location to any of the genes described. This does not detract from the value of the manuscript but I think that this limitation should be acknowledged.

Modifications were made to this section.

It is not really surprising that corals possess environmental sensors in their genomes. This would be true of just about all eukaryote cells. To infer that the presence of sensing components (at the end of the subsection “Environmental and stress response systems”) that are pretty ubiquitous indicates a special propensity to respond to varying environmental signals is not really original. More interesting would be the demonstration of further novel environmental signalling components, as indicated by the analysis of circadian genes and the interesting findings with extrinsic and intrinsic apoptotic pathways. In relation to the latter, a better explanation of intrinsic and extrinsic pathways would add clarity to the Discussion.

Modifications were made to this section.